# Model of Random Field with Piece-Constant Values and Sampling-Restoration Algorithm of Its Realizations

**DOI:** 10.3390/e21080792

**Published:** 2019-08-14

**Authors:** Yuri Goritskiy, Vladimir Kazakov, Olga Shevchenko, Francisco Mendoza

**Affiliations:** 1Department, Moscow Power Engineering Institute, Technical University, Krasnokazarmennya 14, 111250 Moscow, Russia; 2National Polytechnic Institute of Mexico, Ave. IPN s/n, Building Z, Access 4, 3th Floor, SEPI-Telecommunications, 07738 Mexico City, Mexico

**Keywords:** non-gaussian model of a random field with an arbitrary number of states, sampling-reconstruction procedure of such model, reconstruction algorithm, reconstruction error algorithm

## Abstract

We propose a description of the model of a random piecewise constant field formed by the sum of realizations of two Markov processes with an arbitrary number of states and defined along mutually perpendicular axes. The number of field quantization levels can be arbitrary. Realizations of a random field model of the desired shape are created by appropriate selection of parameters for formative realization of Markov processes. For the proposed field model, we investigated the sampling and restoration algorithm of any selected realizations. As a result, we determined the optimal sampling and recovery algorithms. The resulting sampling is fundamentally non-periodic. Recovery errors are calculated. Two examples are considered.

## 1. Introduction

In image processing, the first operation performed on a specific plot in most cases is level quantization. The number of levels depending on the objectives of the study may be different. The second operation is to sampling of the quantized plot using Cartesian (or polar) coordinates. The problems that arise here are typical for the sampling-recovery algorithms (SRA) of realizations of random fields; it is necessary to determine based on a priori information about the image and selected criteria, the optimal sampling-recovery procedure and then evaluate the quality of restoration.

Gaussian models of random fields are the simplest, because for multidimensional distributions of random variables, the analytical expressions representing a priori information about the field are known. The situation with non-Gaussian field models is fundamentally different, since the analytical description becomes very complicated and the level quantization operation only aggravates the situation. This circumstance explains the fact that the number of real images being analyzed is constantly growing, see [1,2,3] etc.

Various models of the random fields, the realizations of which are quantized by level and sampled on a plane, are known in the literature (see, for example, the fundamental books [4,5,6,7,8], as well as the journal articles [9,10,11,12,13,14]. However, the authors of the present article are not aware of any works discussing the statistical models of random non-Gaussian fields with a finite number of states (i.e., analytical field models after a quantization). In addition, there are no publications devoted to the SRA description of realizations of such fields. The proposed work is intended to remedy the aforementioned shortcomings.

The first part of this work deals with a generalization of a random field model with jumps (or piecewise constant states). The first publication devoted to the analytical description of non-Gaussian fields with jumps in brightness is [15], which is a “chessboard with random rectangles”. This model was further generalized in [16,17]. In [17] the number of levels was increased to four. Realizations of such random fields with piecewise constant states are produced by the summation of realizations of binary Markov processes given along the coordinate axes. In the present work we generalize the model [17] by forming the field realizations as a sum of realizations of piecewise constant Markov processes with an arbitrary number of states (such processes are called markovian chains with continuous time). Obviously, field realizations also have an arbitrary number of states. Using realizations of Markov processes as generators of field realizations makes it possible to analytically describe a random field model with piecewise constant states.

The second part of the work is devoted to the study of the sampling-recovery algorithm (SRA) of the realizations of the proposed field model. The SRA for random continuous processes and fields is widely known. However, the studies of SRA of realizations of discontinuous processes and fields are scarce. We begin with a brief discussion of SRA of random processes with piecewise constant states. SRA for the realizations of such processes have pronounced differences from features SRA for realizations of continuous random processes. Namely, in case of SRA of continuous processes, it is necessary to determine the recovery function in the interpolation and/or extrapolation modes, as well as assess the quality of the recovery. In that case the recovery function and the recovery error are the functions of time. SRA for realizations of piecewise continuous processes are completely different, since here it becomes necessary to determine the procedure for estimating the instants of transitions from one state to another, as well as assess the estimation´s the quality. In other words, it comes down to estimating a set of random variables. Naturally, in both cases SRA are based on a priori information about random processes, which are significantly different.

Realizations of random processes and fields in SRA are investigated by applying the conditional mean rule (CMR). The optimal structures of the reducing agents of realizations are determined for an arbitrary number and location of the samples. The conditional mean recovery algorithm automatically provides the minimum mean square recovery error. Numerous examples of the application of the discussed technique can be found in [18,19,20,21]. In particular, on the basis of this method were analyzed the realizations of the SRA binary Markov process [22] and the realizations of the SRA of a piecewise continuous Markov process with an arbitrary number of states [23]. The results of [22] served as the basis for the study of SRA for random piecewise constant fields [16,17] with the number of states equal to 2, 3, 4. In this work we study using the approach [23] the SRA for the field realizations with an arbitrary number of states. It is clear that an increase in the number of levels of field values increases the possibilities of describing random field models quantized by levels. We emphasize that the optimal SRA entails the non-periodic sampling of realizations. This nontrivial feature is overcome by the CMR method used.

The proposed model is flexible, since a specific type of realizations of a field model with the desired features can be formed by an expedient choice of probabilistic characteristics of the generating Markov processes.

The article consists of five sections. The first describes the proposed random field model with piecewise constant states, the second describes the sampling-recovery algorithm of field realizations and discusses the conditions under which realizations of generating processes can be restored using a known field realization. The third one justifies the choice of sampling intervals for realizations of processes for a given probability of state missing. In the fourth section, the instants of transition from one state to another are estimated from discrete samples. The fifth gives two examples that illustrate the proposed algorithms.

## 2. Description of the Model of a Random Field with Piecewise Constant States

We form a random piecewise-constant field ζ(t,s) by summing two independent homogeneous Markov processes ξ(t) and η(t)
(1)ς(t,s)=ξ(t)+η(s)

The processes ξ(t) and η(s) have continuous arguments and discrete sets of states (let it be the sets $X=\left\{x_{i},i=1,2,\ldots ,N_{\xi }-1\right\}$ and $Y=\left\{y_{j},j=1,2,\ldots ,N_{\eta }-1\right\}$ which are defined on the axes t and s. Hereinafter, it is convenient to perceive t and s as time axes. $N_{\xi },N_{\eta }$ are the numbers of states of the corresponding processes. We do not introduce special notation for realizations of random functions that are different ξ(t),η(t) and ζ(t), therefore Equation (1) will also be understood as the sum of realizations. The Markov property of formative processes allows an analytic description of the structure of such a field. It might be possible to abandon the Markov properties of the generating processes, but this significantly complicates the analytical description of the SRA (for comparison, see, for example, [24]).

It is assumed that the characteristics of the processes ξ(t) and η(s) (the output density and the transition probability) are known. Denote the output density (or output intensity) of the states: for $\xi \left(t\right):\lambda_{\xi i},i=1,2,\ldots ,N_{\xi }-1$, for $\eta \left(s\right):\lambda_{\eta j},j=1,2,\ldots ,N_{\eta }-1$ the meaning of which is tat:$$P\left\{\mathrm{go}\;\mathrm{out}\;\mathrm{from}\;\mathrm{the}\;\mathrm{state}\;x_{i}\;\mathrm{during}\;\mathrm{the}\;\mathrm{interval}\;\Delta t\right\}=\lambda_{\xi i}\Delta t+o\left(\Delta t\right)$$
$$P\left\{\mathrm{go}\;\mathrm{out}\;\mathrm{from}\;\mathrm{the}\;\mathrm{state}\;y_{j}\;\mathrm{during}\;\mathrm{the}\;\mathrm{interval}\;\Delta s\right\}=\lambda_{\eta j}\Delta s+o\left(\Delta s\right)$$

We also assume that for ξ(t) and η(s) there are given graphs or transition probability matrices $P_{\xi }=\left[p_{ik}\right],P_{\eta }=\left[q_{js}\right]$.

Figure 1 shows examples of realizations of fields formed by the sum Equation (1). The number of levels according to the values of each function in Figure 1a–c is 10 (a total of 19 field levels). High intensity of red color means the maximum value of the field, and high intensity of blue color minimum.

Figure 1a–c illustrate the flexibility of the proposed model. It is possible to form various realizations of the random field by selecting the parameters of the generating realizations. Specifically, it is possible to create field realizations with a fairly large area where there are no transitions. Such areas can be in the center (Figure 1a) or shifted in any direction (Figure 1b shows an upward). It is also possible to simulate two maxima (Figure 1c). Small areas with different sizes can be created around such areas, and so on. The Figure 2 shows a field with a maximum at the center (similar to Figure 1a) with a coarser quantification 8 levels for field values.

## 3. Description of the Sample-Recovery Algorithm of the Realization of the Field Model

Unrecoverable errors associated with the omission of states may occur in the sampling of realizations of processes with piecewise constant states. Such errors occur in the case when the duration of the process in the state is less than the sampling interval. The SRA of realization of the processes under consideration should take into account this phenomenon. Namely, it is necessary to determine the SRA so that when recovering, the probability of skipping a state is no more than a given level γ [22,23]:
(2)P{skiping of state}≤γ

This post expresses the idea of the condition: “it is advisable not to skip states”. A similar requirement must be met when analyzing the SRA of a random field ζ(t,s). Below, in Section 4, we detail the condition Equation (2) is detailed.

The description of the SRA realizations of the field ζ(t,s) is associated with the analysis of the SRA realizations of the generating processes ξ(t) and η(s). At the same time, knowing the samples of the field realization, one must not only estimate the position of the switching points of the process realizations, but also determine the states of the processes themselves. The latter circumstance is important because the residence time of the process in a given state and the sampling interval depend on the state itself. We emphasize this circumstance, since sampling turns out to be fundamentally non-periodic.

To determine the realizations of ξ(t) and η(s) from known values of the field ζ(t,s), it is enough to know its value at one point of one of the formative realizations (ξ(t) or η(s)). For example, if at some point s* the value η*=η(s*) is known, then by Equation (1) we have
ξ(t)=ζ(t,s*)−η(s*)=ζ(t,s*)−η*
(3)ξ(t)=ζ(t,s*)−η(s*)=ζ(t,s*)−η*.

Then at an arbitrary point t* of the t axis one can write:
ξ(t*)=ζ(t*,s*)−η(s*)=ζ(t*,s*)−η*

Finally, the application Equation (1) yields
(4)η(s)=ζ(t*,s)−ξ(t*)=ζ(t*,s)−ξ*.

The possibility of unambiguous recovery depends on the sets of states of the formative processes and on the specific states used in a given field realization. There may be situations where recovery is trivial. For example, let the process ξ(t) have d states: $x_{i}=i\Delta ,i=1,2,\ldots ,d-1$ (Δ is the measurement discreteness; we take Δ=1) and the process η(s) has M states: $y_{j}=j\Delta ,j=0,1,2,\ldots ,M-1$.

Table 1 shows the values of the function z(x,y)=x+y for arbitrary d and M.

In particular, for d=M=4, it turns out dM=16 field levels. The values of this function are given in Table 2.

In the examples, all values of the function z=f(x,y)=x+y are different, and therefore each field value corresponds to a single pair (x,y), so restoring the realizations of the processes ξ(t) and η(s) is trivial.

The situation when the process state grids have the same step is quite important. For example, let the processes ξ(t) and η(s) have N integer states:
$x_{i}=i,y_{j}=j;i,j=1,2,\ldots ,N-1$. The values of the sum with N=4 are shown in Table 3. It is clear that if the field has the values 1, 2, 3 or 5, then the value of the pair (x,y) is not uniquely determined. If the value of the field is 0 or 6, then the value of the pair (x,y) is uniquely determined, and therefore the realizations ξ(t) and η(s) are determined in accordance with Equations (3) and (4).

For this important case (the case of sets with the same grid step), the necessary and sufficient condition for the uniqueness of the recovery of generating realizations is the following statement: among the values of the realization of the field ζ(t,s) there is either the minimum possible value
$$\underset{t,s}{\min}\zeta \left(\xi \left(t\right),\eta \left(s\right)\right)=\underset{\xi ,\eta }{\min}\zeta \left(\xi ,\eta \right)=\zeta_{\min}=x_{\min}+y_{\min}=z_{\min}$$
(5)$$\underset{t,s}{\min}\zeta \left(\xi \left(t\right),\eta \left(s\right)\right)=\underset{\xi ,\eta }{\min}\zeta \left(\xi ,\eta \right)=\zeta_{\min}=x_{\min}+y_{\min}=z_{\min}$$
or the maximum possible value
(6)$$\underset{t,s}{\max}\zeta \left(t,s\right)=\underset{\xi \eta }{\max}\zeta \left(\xi ,\eta \right)=\zeta_{\max}=x_{\max}+y_{\max}=z_{\max}$$
where $\left(x_{\min},y_{\min}\right)$, $\left(x_{\max},y_{\max}\right)$ are the minimum and maximum values of the sets X and Y.

The sufficiency is obvious, since each of these values corresponds to a single pair: $\left(x_{\min},y_{\min}\right)$ for $\zeta_{\min}$, and $\left(x_{\max},y_{\max}\right)$ for $\zeta_{\max}$, and then by Equations (3) and (4) the realizations ξ(t) and η(s) are restored.

The necessity: from the uniqueness follows the fulfillment of condition Equations (5) and (6). Suppose that none of the conditions Equations (5) and (6) is satisfied, i.e., among the values (ξ,η) there is neither a point $\left(x_{\min,}y_{\min}\right)$, nor a point $\left(x_{\max},y_{\max}\right)$. Or else, this condition is written differently: for any value (ξ,η)=(x,y),
(7)$$z_{\min}+1\leq \zeta \left(\xi ,\eta \right)=x+y\leq z_{\max}-1$$
and show that recovery is not unique. Indeed, the available values of z=ζ(ξ,η) fields are obtained by summing some values of $x_{i}$ and some values of $y_{j}$, and therefore the system of algebraic equations
(8)$$\zeta_{i,j}=x_{i}+y_{j}$$
relatively unknown terms $x_{1},x_{2},\ldots ,y_{1},y_{2},\ldots$ is solvable by construction (the question is whether the solution is unique). Let be $x_{1},x_{2},\ldots ,y_{1},y_{2},\ldots$ that generate the available values of $\zeta_{ij}$ field.

Condition Equation (8) (the absence of two “necessary” points $\left(x_{\max},y_{\max}\right)$ and $\left(x_{\min},y_{\min}\right)$) is satisfied only in two cases:$$\mathrm{all}\;x_{i}\geq x_{\min}+1\;\mathrm{and}\;\mathrm{all}\;y_{j}\leq y_{\max}-1,$$
(9)$$\mathrm{all}\;x_{i}\geq x_{\min}+1\;\mathrm{and}\;\mathrm{all}\;y_{j}\leq y_{\max}-1,$$
(10)$$\mathrm{or}\;\mathrm{all}\;x_{i}\leq x_{\max}-1\;\mathrm{and}\;\mathrm{all}\;y_{j}\geq y_{\min}-1.$$

These two situations are illustrated in Figure 3a,b.

We form a new solution by marking the values with strokes. If Equation (9) is true, then from each $x_{i}$ we subtract 1, and to each $y_{j}$ we add 1:
$$x_{i}^{\prime }=x_{i}^{\prime }-1,\;y_{j}=y_{j}+1.$$

It is clear that $x_{1}^{\prime },x_{2}^{\prime },\ldots ,y_{1}^{\prime },y_{2}^{\prime },\ldots$ also satisfies system Equation (8), and in the Figure 3, the range of values a) goes to region b).

If true Equation (10), then we add to each $x_{i}$ 1 and subtract 1 from each $y_{j}:x_{i}=x_{i}+1,y_{j}=y_{j}-1.$

These values also satisfy system Equation (8), while in Figure 3 the domain of (b) goes into region (a). Thus, if condition Equations (5) and (6) is not satisfied, then the solution is not unique. The necessity is shown.

Note that if the uniqueness condition is not satisfied, then the restoration is possible up to constants, and this is quite acceptable in many practical situations.

So, the necessary and sufficient condition for the uniqueness of the recovery of formative realizations ξ(t) and η(s) is the existence (among the values of a given field realization) of an least one extreme value $\zeta *(\zeta_{\max}\mathrm{or}\;\zeta_{\min})$ of a field that can be represented by the sum in the only way:
(11)ζ*=ξ*+η*, ξ*∈X, η*∈Y
where X and Y are the sets of the states of the processes, respectively, ξ(t) and η(s). 

The sampling procedure consists of the following:(1)Determining the values ζ*,ξ*,η* among the given field values together with the corresponding values of the arguments t* and s*.(2)The definition of formative realizations ξ(t) and η(s) by Equations (3) and (4).(3)Calculation of sampling intervals $T_{n}^{\xi }$ and $T_{m}^{\eta }$ along the t and s axes and sampling points: $t_{n+1}=t_{n}+T_{n}^{\xi },s_{m+1}=s_{m}+T_{m}^{\eta },t_{0}=0,s_{0}=0,\;n,m=1,2,\ldots$.(4)Measurement of values
(12)$$\xi_{n}=\zeta \left(t_{n},s*\right)-\eta *,\;\eta_{m}=\zeta \left(t*,s_{m}\right)-\xi^{*}$$

The sampling results are sets of values
$$\left(t_{0},\xi_{0}\right),\left(t_{1},\xi_{1}\right),\ldots ,\left(t_{n},\xi_{n}\right),\ldots$$
$$\left(s_{0},\eta_{0}\right),\left(s_{1},\eta_{1}\right),\ldots ,\left(s_{m},\eta_{m}\right).$$

The field recovery procedure consists of the following actions:
(1)Restoration of realizations $\xi_{v}\left(t\right)$ and $\eta_{v}\left(s\right)$ by the values of $\xi_{n}$ and $\eta_{m},n,m=1,2,\ldots ;$(2)Determining the recovered values of the $\zeta_{v}\left(t,s\right)$ field in accordance with Equation (1):
$$\zeta_{v}\left(t,s\right)=\xi_{v}\left(t\right)+\eta_{v}\left(s\right)$$

## 4. Determination of Sampling Intervals

SRA for realization of Markov processes with a finite number of states are described in [22]. Let a continuous time Markov process have states 0,1,2,…,N−1 with the corresponding exit from states intensities $\lambda_{0},\ldots ,\lambda_{N-1}$ and the matrix of conditional probabilities $P=\left[p_{ij}\right]$ (on the diagonal $p_{ii}=0$). Let $\theta_{i}$ be the duration of being in state i, and $\theta_{j}$ be the duration of being in the next state j. By the property of a Markov process, the random variables $\theta_{i}$ and $\theta_{j}$ are independent and distributed according to the exponential law with the parameters $\lambda_{i}$ and $\lambda_{j}$ with density
$$f\left(x;\lambda \right)=\lambda e^{-\lambda x},x\geq 0,\;$$
and with the corresponding distribution function F(x,λ).

At a sampling instant $t_{n}$, it is necessary to determine the next instant $t_{n+1}=t_{n}+T_{n}$, i.e., determine the sampling interval $T_{n}$, which is selected from the condition Equation (2) of a given small probability γ of a gap following the state i unknown state j.

Next, we will use the property of the exponential distribution: the conditional distribution of the remainder of the transition does not depend on the elapsed waiting time and coincides with the distribution of the time in the state. The event “skip of state” (mentioned in Equation (2)) means that we have a condition:
$$P\left\{\theta_{i}+\theta_{j}<T\right\}\leq \gamma$$
for any j, for which $p_{ij}>0$, i.e., the maximum of the probability in j must be equal to γ:
(13)$$\underset{j:p_{ij}>0}{\max}P\left\{\theta_{i}+\theta_{j}<T\right\}=\underset{j:p_{ij}>0}{\max}F_{\sum }\left(T;i,j\right)=F_{\sum }\left(T;i,j\right)=\gamma$$
where $F_{\sum }\left(T;i,j*\right)$ is the distribution function of the sum $\sum =\theta_{i}+\theta_{j}$. Obviously, the greater $\lambda_{j}$ is, the less is $\theta_{j}$. Equation (13) reaches its maximum in that state j*=j*(i), at which $\lambda_{j}$ is maximal:
(14)$$\lambda_{j*}=\underset{j,P_{ij}\neq 0}{\max}\lambda_{j}$$

The distribution function of the sum $\Sigma =\theta_{i}+\theta_{j^{*}}$ is equal to $F_{\Sigma }\left(x;\lambda_{i}.\lambda_{j^{*}}\right)$. Note that $\lambda_{i}$ and $\lambda_{j^{*}}$ enter this expression symmetrically, which allows us to move to a one-parameter family of functions. Denote
(15)$$\alpha =\max\left(\lambda_{i},\lambda_{j*}\right),\beta =\min\left(\lambda_{i},\lambda_{j*}\right),k=\beta /\alpha ,0<k\leq 1$$

It is easy to show, the probability density of a random variable α∑ is determined by formula:$$p_{\alpha \sum }\left(t\right)=\frac{1}{\alpha }p_{\sum }\left(t/\alpha \right)=\left\{ \begin{array}{c} \left(e^{-kt}-e^{-t}\right)k/\left(1-k\right),\;if\;k<1\\ te^{-t},\;if\;k=1 \end{array}\right..$$

Then for a random variable αΣ, the distribution function, depending on the parameter k, as is easily shown [23], is equal top
(16)$$F_{\alpha \Sigma }(x,k)=\{\begin{array}{c} 1-\left(e^{-kx}/k-e^{-x}\right)k/\left(1-k\right),\;\mathrm{if}\;k\neq 1,\\ 1-(1+x)e^{-x},\;\mathrm{if}\;k=1. \end{array}$$

Condition Equation (13) takes the form:(17)$$P\left(\theta_{i}+\theta_{j*}<T\right)=P\left[\alpha \left(\theta_{i}+\theta_{j*}\right)<\alpha T\right]=F_{\alpha \sum }\left(\alpha T,k\right)=\gamma$$

From Equations (16) and (17), one can find the value of the sampling interval:(18)$$T=T\left(i,\gamma \right)=\frac{1}{\alpha }F_{\alpha \sum }^{-1}\left(\gamma ,k\right)$$
where $F_{\alpha \sum }^{-1}\left(\gamma ,k\right)$ is the function inverse to $F_{\alpha \sum }\left(\alpha T,k\right)$ with respect to the first variable (the monotonic increase in x of the function $F_{\alpha \sum }\left(\alpha T,k\right)$ is taken into account). Note that the dependence of the interval T on i appears through α and k. The graphs of functions $F_{\alpha \sum }^{-1}\left(\gamma ,k\right)$ [23] depending on the parameter k for different γ are shown in Figure 4. There is also an approximate solution of Equation (16) for T obtained by expanding in powers of small x [23]:
(19)$$T\approx \sqrt{2\gamma /\alpha \beta }$$

The quality of the approximation Equation (19) with respect to the exact solution is no worse than 10% at γ<0.03, [23].

Let at the sampling instant $t_{n}$ the sample to have the value
$$\xi \left(t_{n}\right)=\xi_{n}=x_{i}.$$

The least favorable next state, according to Equation (14), is the state with the number j*=j*(i). We calculate the quantities α,β and k by the relation Equation (17), and the value $T_{n}^{\xi }=T\left(i,\gamma \right)$ is selected by the Equation (18). Then the next sampling time is determined in an obvious way
$$t_{n+1}=t_{n}+T_{n}^{\xi }.$$

Note that the functions j*=j*(i) and T(i,γ) are calculated in advance. The sampling of η(s) is performed similarly. Sampling results are sequences of pairs $\left(t_{n},\xi_{n}\right),n=0,1,2,\ldots$ and similarly $\left(s_{m},\eta_{m}\right),m=0,1,2,\ldots$

## 5. Results Recovery of Formative Realizations from Discrete Samples

Consider the realization of the process on the axis (0,t) and denote on it the sampling points that need to be determined: $t_{1},\ldots ,t_{n},t_{n+1},\ldots ;t_{n+1}=t_{n}+T_{n}^{\xi },t_{0}=0$. Suppose that on the sampling interval $\left(t_{n}+T_{n}^{\xi }\right)$ there is a transition from state i to state j: $\xi \left(t_{n}\right)=i\neq \xi \left(t_{n}+T\right)=j$ (here $T=T_{n}^{\xi }$).

It is required to estimate the unobservable value τ-the random istant of transition from i to j. The best estimate (in terms of minimal error variance) is the expectation of conditional distribution τ, provided that $\tau \in \left(t_{n},t_{n}+T\right)$. Due to the homogeneity of the process, we assume that τ∈(0,T). The density of the conditional distribution of the point τ on the interval (0,T) provided that there is only one transition in the interval, is determined by the following relation [23]:
(20)pτ(x|i,j)={(μξe−μξx)/(1−e−μξx), if μξ≠0,1T,  if  μξ=0,
where $\mu_{\xi }\equiv \mu_{\xi }\left(i,j\right)=\lambda_{\xi_{i}}-\lambda_{\xi_{j}}$. Equation (20) describes a truncated exponential distribution, which in the case $\mu_{\xi }=0$ degenerates into a uniform one. By density Equation (20), the conditional expectation is determined, which is an estimate $\overset{⌢}{\tau }$ with the minimum variance for the transition instant τ:
(21)τ^ij=E{τ|i,j}={μξ−1(1−μξT/(eμξT))=T[0.5+g(u)],μξ≠0,T2,μξ=0
where indicated: $u=\mu_{\xi }\left(i,j\right)T$,
$$g\left(u\right)=u^{-1}\left(1-u/\left(e^{u}-1\right)\right)-0.5=u^{-1}+0.5\left(1+e^{u}\right)/\left(1-e^{u}\right),$$

We can verify that g(u) is an odd function and g(0)=0.

Specifically,
(22)$${\hat{\tau }}_{ij}\equiv E\left\{\tau |i,j\right\}=T\left[0.5+g\left(u_{ij}\right)\right],\;u_{ij}=\left(\lambda_{\xi_{i}}-\lambda_{\xi_{j}}\right)T=\mu_{\xi }T$$
the expression for the variance of the estimate is
(23)$$V{\hat{\tau }}_{ij}=\{\begin{array}{c} \frac{1}{\mu_{\xi }^{2}}\left[1-{\left(\mu_{\xi }T\right)}^{2}/\left(e^{\mu_{\xi }T}+e^{-\mu_{\xi }T}-2\right)\right],\;\mu_{\xi }\neq 0,\\ T^{2}/12,\;\mu_{\xi }=0, \end{array},i\neq j$$

Graphs of the relative values of the estimate $\hat{\tau }/T=\left[0.5+g\left(u\right)\right]$ and the standard deviation $\sigma =\sqrt{V\hat{\tau }}/T$, as functions of the argument u=μT are shown in Figure 5.

Using Equation (22), we obtain the estimated values for the process switching points. The restoration is performed according to the values $\xi_{n},\eta_{m};n,m=1,2,\ldots ,$ measured in accordance with Equation (12).

For the processes ξ(t): if the interval $\left(t_{n},t_{n+1}\right)$ is the switching interval, i.e., $\xi_{n}\neq \xi_{n+1}$, then we determine the value of the estimate of the transition point ${\overset{⌢}{\vartheta }}_{k}^{t}$ using the Equation (22):$${\overset{⌢}{\vartheta }}_{k}^{t}=t_{n}+{\overset{⌢}{\tau }}_{ij},$$
where k is the sequence number of the switching interval on the t axis, k=1,2,… The value of the recovered realization $\xi_{v}\left(t\right)$ at the point t of the next interval is determined by the relation: $\xi_{v}\left(t\right)=\xi_{n}$ if $t\in [{\hat{\vartheta }}_{k-1}^{t},{\hat{\vartheta }}_{k}^{t}),\;{\hat{\vartheta }}_{0}^{t}=0$. 

For the process η(s): if the interval $\left(s_{m},s_{m+1}\right)$ is the next transition interval, i.e., $\eta_{m}\neq \eta_{m+1}$, then we determine the value of the transition point ${\hat{\vartheta }}_{k}^{s}$ estimate using Equation (22):(24)$${\hat{\vartheta }}_{k}^{s}=s_{m}+{\hat{\tau }}_{ij}$$
in Equation (22), instead $\mu_{\xi }$ we use $\mu_{\eta }\equiv \mu_{\eta }(i,j)=\lambda_{\eta i}-\lambda_{\eta j}$, i≠j; instead $T=T^{\xi }$ we use $T=T^{\eta }$; k is the sequence number of the transition interval on the s axis, k=1,2,… The values of the reconstructed realization at points s are as follows:
(25)$$\eta_{v}\left(s\right)=\eta_{m}\;\mathrm{if}\;s\in [{\hat{\vartheta }}_{k-1}^{s},{\hat{\vartheta }}_{k}^{s})\;,\;{\hat{\vartheta }}_{0}^{s}=0$$

For the recovered realizations of the processes $\xi_{v}\left(t\right)$ and $\eta_{v}\left(s\right)$ we determine the reconstructed value of the field realization by Equation (1):$$\zeta_{v}\left(t,s\right)=\xi_{v}\left(t\right)+\eta_{v}\left(s\right).$$

## 6. Examples

### 6.1. Example 1. No Skip of State When Sampling

Let the process ξ(t) have 4 states: 0, 1, 2, 3, the process η(t) − 5 states: 0, 1, 2, 3, 4. The transition graph of the process ξ(t) is shown in Figure 6. These arrows show transitions in a way to indication of density. It is assumed that for internal states "1" and "2" transitions are possible only in neighboring states, and with equal probabilities. 

The process graph η(t) differs only in the number of states and values of the output densities. The density of exit from the states and the average duration of stay in the states are shown in Table 4. 

The sum z=x+y can take 8 integer values from 0 to 7. If it is 0 or 7, then the pairs are determined uniquely: for z=0 it is (0, 0), for z=7 it is (3, 4). Figure 7 shows the original field on a 10 × 10 square with 8 color levels. Vertically, there are 5 intervals of constancy, and horizontally-7; total 5 × 7 = 35 rectangles for which the field values, are given in the matrix Z with a size of 5 × 7.
$$Z=\left[\begin{array}{ccccccc} 4 & 5 & 6 & 7 & 6 & 5 & 6\\ 3 & 4 & 5 & 6 & 5 & 4 & 5\\ 2 & 3 & 4 & 5 & 4 & 3 & 4\\ 1 & 2 & 3 & 4 & 3 & 2 & 3\\ 0 & 1 & 2 & 3 & 2 & 1 & 2 \end{array}\right]$$

Among the field values we find the value $z_{\max}=7$, i.e., $x_{\max}=3,y_{\max}=4$ at the point $t^{*}=5,s^{*}=9$, and also one value $z_{\min}=0$ at the point $t^{*}=1,s^{*}=1$. To recover ξ(t) and η(s), one point is enough $t^{*}=5,s^{*}=9$ with $z_{\max}=7$, which means that $\xi^{*}\left(t\right)=\xi \left(t^{*}=5\right)=3$ and $\eta^{*}=\eta \left(s^{*}=9\right)=4$. Then by Equations (3) and (4), we have
$$\xi \left(t\right)=\varsigma \left(t,s^{*}\right)-\eta \left(s^{*}\right)=\varsigma \left(t,9\right)-4,$$
(these are the values of the top row of the matrix minus 4), $\eta \left(s\right)=\varsigma \left(t^{*},s\right)-\xi \left(t^{*}\right)=\varsigma \left(5,s\right)-3$ (these are the values from the bottom up to the middle column of the matrix minus 3). Realizations ξ(t) and η(s) are shown in Figure 8. The corresponding state sequences are as follows: for ξ(t)−0,1,2,3,2,1,2; for η(s)=0,1,2,3,4. 

The sampling intervals are obtained by Equations (14), (17) and (18).

For ξ(t), we have the states 0, 1, 2, 3, for which we write out the values of the sampling intervals $T^{\xi }$ for various values of the permissible probability γ of omission (see Equation (2)): when γ=0.01, $T^{\xi }=0.16;\;0.16;\;0.18;\;0.24;$ when γ=0.05, $T^{\xi }\;=0.39;\;0.39;\;0.44;\;0.57.$

For η(s) we have the states: 0, 1, 2, 3, 4, for which the sampling intervals $T^{\eta }$ are different: when $\gamma =0.01\;,\;T^{\eta }=0.3;0.3;0.3;0.3;0.3;$ when $\gamma =0.05\;T^{\xi }=0.71;0.71;0.71;0.71.$

For convenience of illustration the value of the skip probability is γ=0.05, which gives extended sampling intervals. In Figure 9 shows sampling and reconstruction for ξ(t)) and η(s); reconstructed realizations are depicted by dashed lines. Figure 10 shows the restored field. In this example, no state was skipped during sampling.

### 6.2. Example 2. There is a Skip of State When Sampling

Consider an example close to the previous one, which differs from it only by the formative realization ξ(t). The new realization is shown in Figure 11, where it is noticeable that the second left vertical bar is very narrow.

State 1 in the interval (0, 2) is skipped (Figure 12) when performing a sample. 

For this reason one column of rectangles disappears in the restored field realization of Figure 13. Their number became six instead of seven. This is a characteristic error of a special type, which is governed by the choice of value γ.

We draw attention to the fact that the values found for the sampling intervals of the forming realizations turned out to be different for different states.

## 7. Conclusions

A model of a random field is proposed that is formed by summing two formative realizations of piecewise constant Markov processes with an arbitrary number of states. The flexibility of the field model is associated with the possibility of choosing the number of quantization states, as well as the expedient assignment of probabilistic characteristics of two formative realizations to obtain realizations of the field of the desired shape (for example, the presence of relatively large areas without jumps, their location in the field of view, etc.). This article has developed a sampling-restoration algorithm of the proposed random field model. We derive for the field type under consideration, the necessary and sufficient condition for the unambiguous recovery of formative realizations from field values. We substantiate the choice of sampling intervals with a given probability of missing state. Discrete samples are used to estimate random moments of transition from one state to another as well as the variance of the estimates.

The article has developed a sampling-restoration algorithm of the proposed random field model. We obtain for this model the necessary and sufficient condition for the unambiguous recovery of formative realizations from the field´s values. The choice of sampling intervals with a given probability of skipping a state is justified. On discrete samples, random instants of transition from one state to another are estimated, and the estimates of variance are calculated. Two examples are given to illustrate the proposed algorithms.

The proposed model can be used for modeling the various quantized by level fields. It is possible to select the number of quantization levels, as well as the formation of field realizations with the desired features.

## Figures and Tables

**Figure 1 entropy-21-00792-f001:**
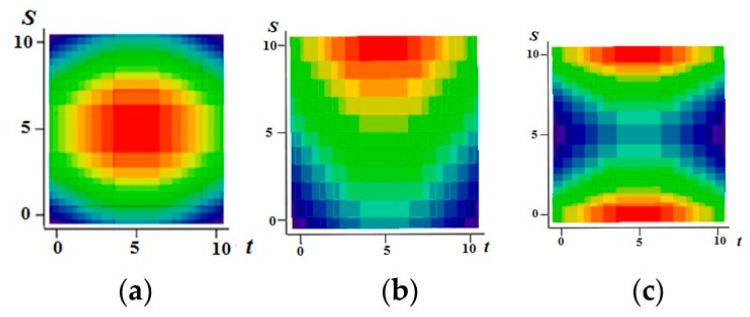
Examples of fields formed by the sum of realizations of two Markov processes. (**a**) In the center there is a rather large area of the field without transitions with the maximum level of quantization; (**b**) Here, this part of the model is shifted upwards; (**c**) Parts of the realization of the high quantization field model are separated and located in the upper and lower regions.

**Figure 2 entropy-21-00792-f002:**
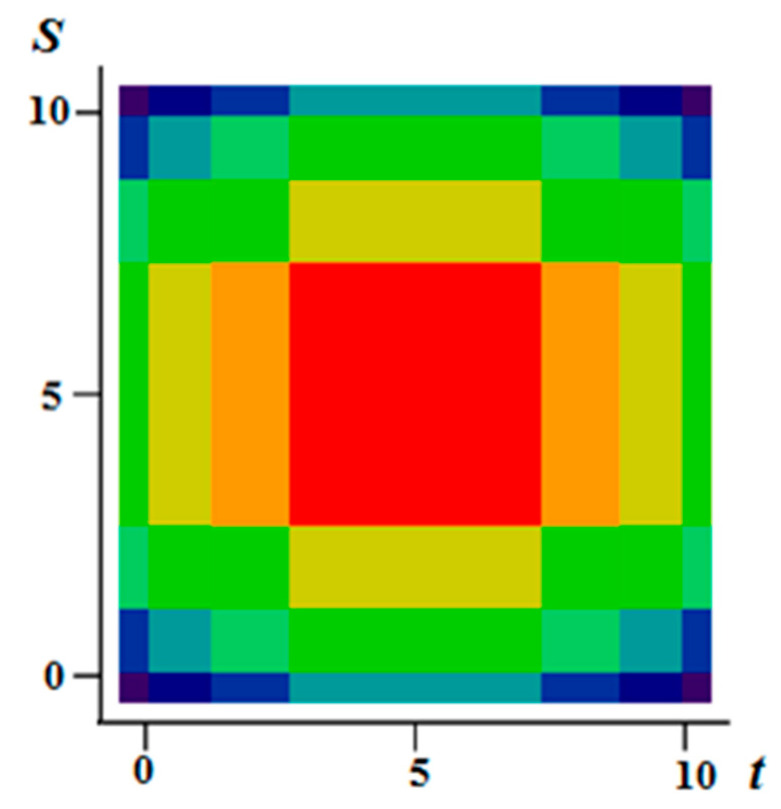
A field with a pronounced maximum and with a coarse discretization of eight levels.

**Figure 3 entropy-21-00792-f003:**
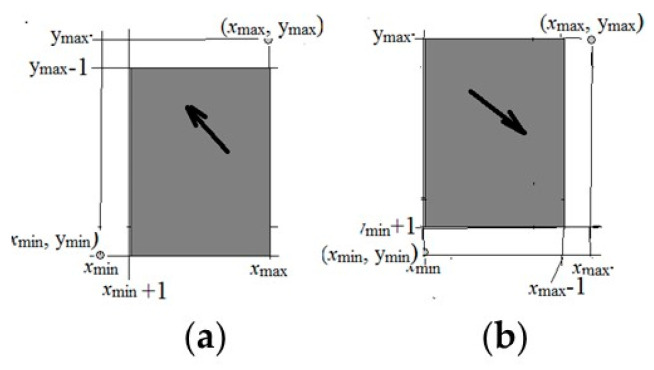
The range of values (x,y) for conditions Equation (7) is shaded, where $x_{\max}$ and $y_{\max}$ are maximum values, $x_{\min}$ and $y_{\min}$ are minimal values. (**a**) A case illustrating condition Equation (9); (**b**) A case illustrating condition Equation (10).

**Figure 4 entropy-21-00792-f004:**
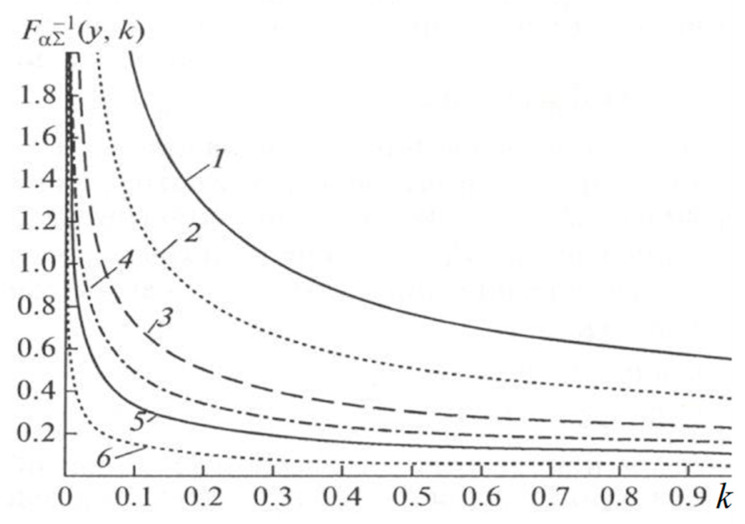
Function $F_{\alpha \Sigma }^{-1}\left(\gamma ,k\right)$ for different values γ: 0.1(1), 0.05(2), 0.02(3), 0.01(4), 0.005(5), 0.001(6).

**Figure 5 entropy-21-00792-f005:**
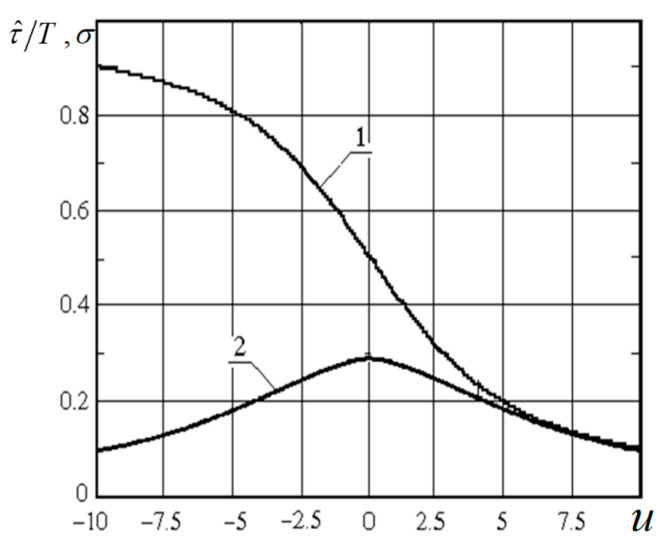
Relative evaluation values $\hat{\tau }/T$ (curve 1) and its standard deviation $\sigma =\sqrt{V\hat{\tau }}/T$ (curve 2) as a function of the parameter u=μT.

**Figure 6 entropy-21-00792-f006:**
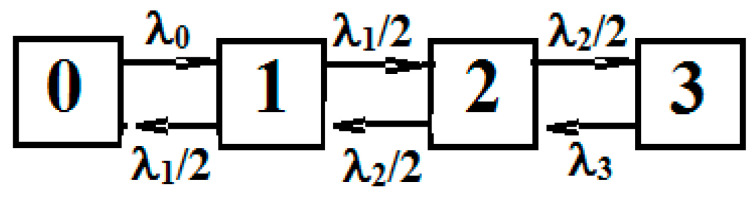
Graph flowchart of the process ξ(*t*).

**Figure 7 entropy-21-00792-f007:**
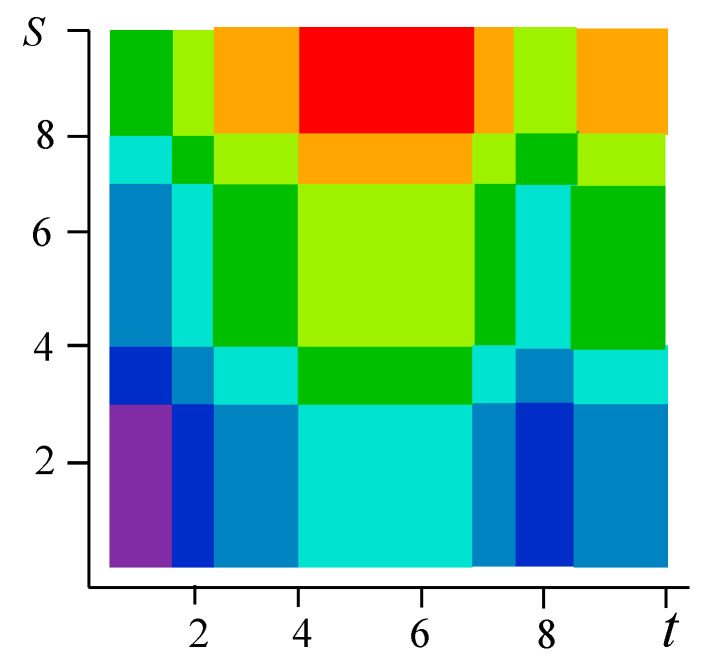
The specified realization of the field (Example 1).

**Figure 8 entropy-21-00792-f008:**
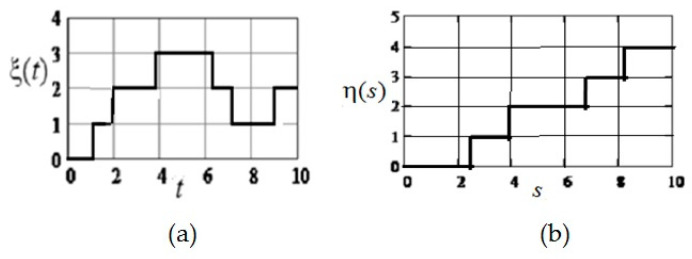
Realizations that make up a field (Example 1). (**a**) Realization of the forming Markov process, fixed along the axis ξ(t); (**b**) Realization of the forming Markov process, fixed along the axis η(t).

**Figure 9 entropy-21-00792-f009:**
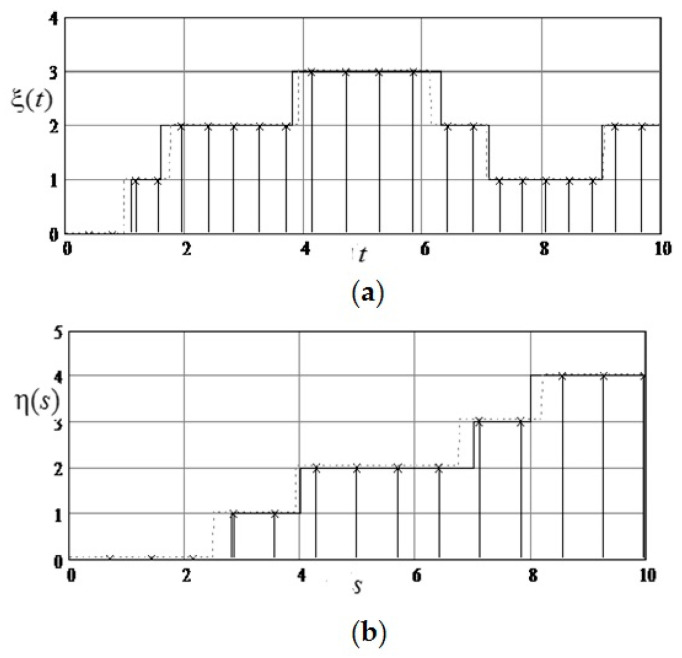
Sampling and recovery of formative realizations (Example 1). (**a**) Sampling and recovery of formative realization of Markov process, fixed along the axis ξ(t); (**b**) Sampling and recovery of formative realization of Markov process, fixed along the axis η(s).

**Figure 10 entropy-21-00792-f010:**
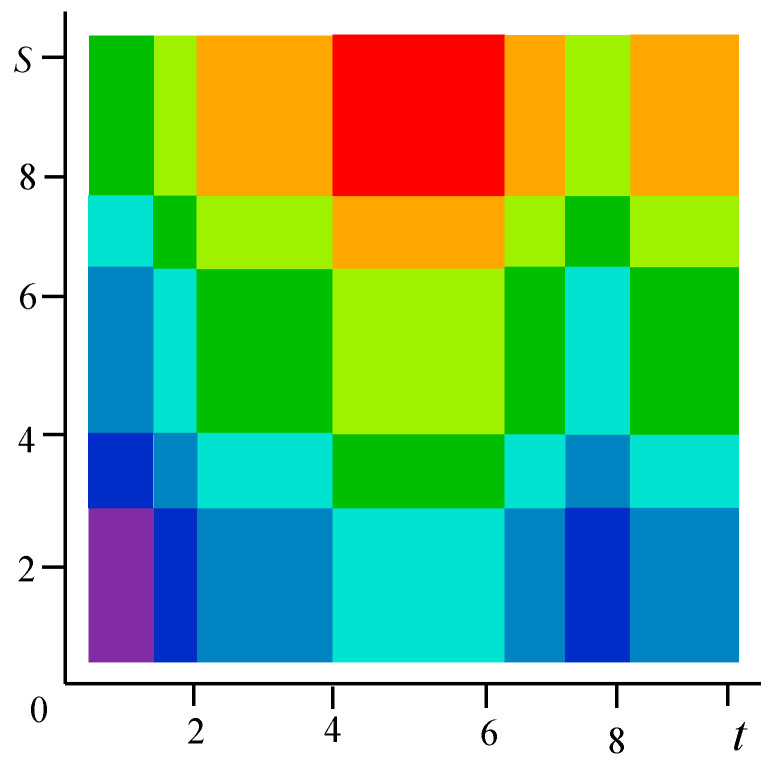
The recovered field (Example 1).

**Figure 11 entropy-21-00792-f011:**
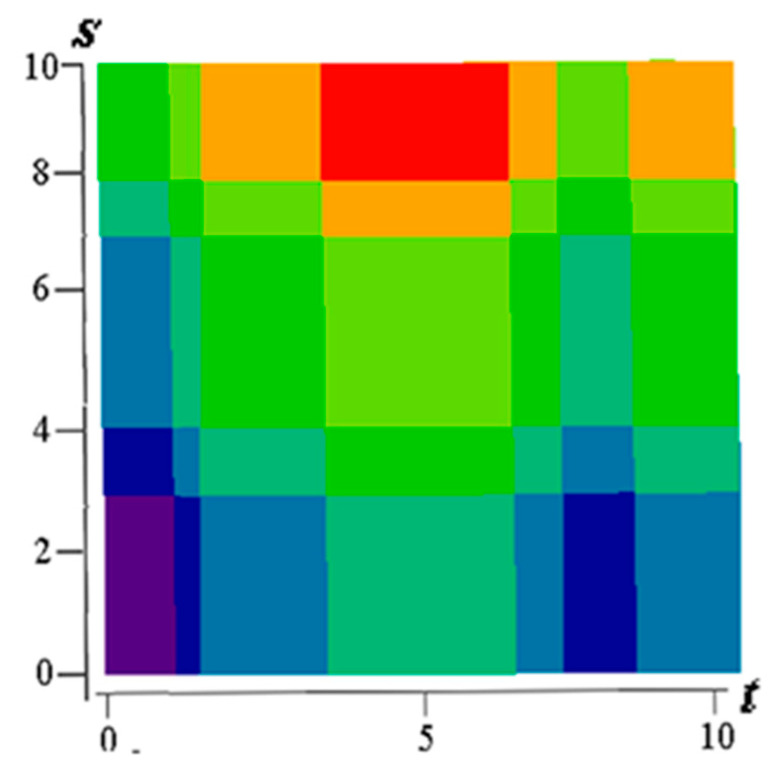
Specified field realization (Example 2).

**Figure 12 entropy-21-00792-f012:**
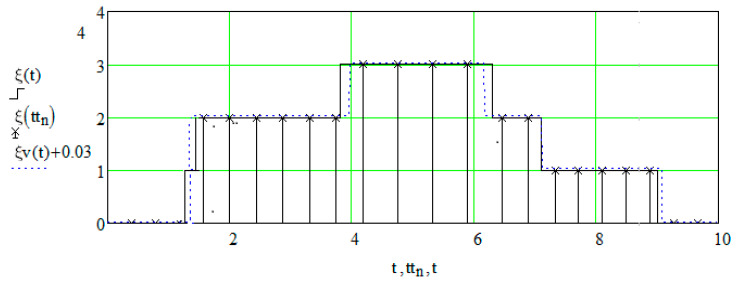
Discretization and recovery realization of ξ(t) (Example 2).

**Figure 13 entropy-21-00792-f013:**
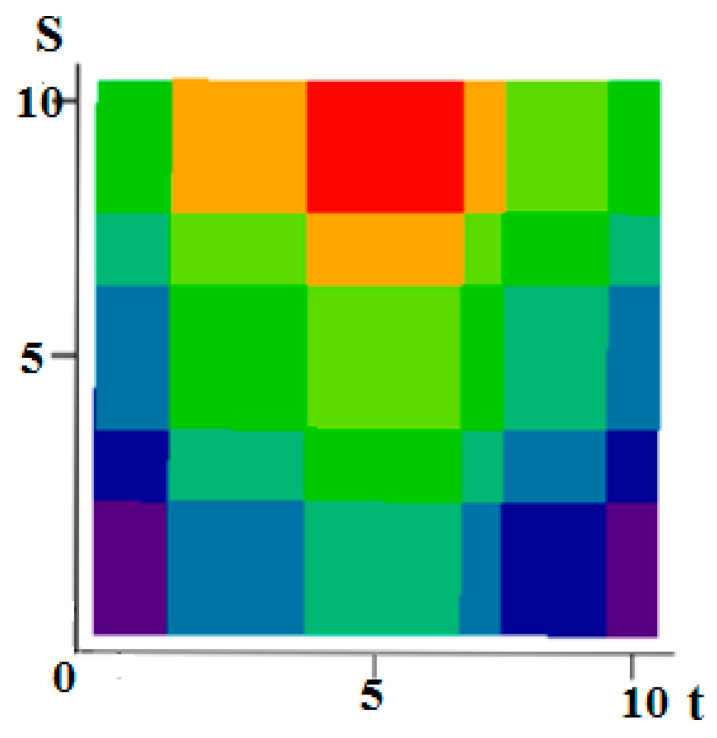
Recovered field (Example 2).

**Table 1 entropy-21-00792-t001:** Values z = *f*(*x*,*y*) = *x* + *y* for arbitrary *d* and *M*, total *N* = *dM* values.

*z* = *x* + *y*		*y*				
		0	*d*	2*d*	…	(*M* − 1)*d*
*x*	0	0	0 + *M*	0 + 2 *M*	…	0 + (*M* − 1)*d*
	1	1	1 + *M*	1 + 2 *M*	…	1 + (*M* − 1)*d*
	2	2	2 + *M*	2 + 2 *M*	…	2 + (*M* − 1)*d*
	⁞	⁞	⁞	⁞	⁞	⁞
	*d* − 1	*d* − 1	2*d* − 1	3*d* − 1	…	*Md* − 1

**Table 2 entropy-21-00792-t002:** Values *z* = *f*(*x*,*y*) = *x* + *y* for *d* = *M* = 4. total *N* = 16 values.

*z* = *x* + *y*		*y*			
		0	4	8	12
*x*	0	0	4	8	12
	1	1	5	9	13
	2	2	6	10	14
	3	3	7	11	15

**Table 3 entropy-21-00792-t003:** Field values at *M* = 4 states.

*z* = *x* + *y*		*y*			
		0	1	2	3
*x*	0	0	1	2	3
	1	1	2	3	4
	2	2	3	4	5
	3	3	4	5	6

**Table 4 entropy-21-00792-t004:** State Characteristics.

	Process ξ (*t*)				Process η (s)				
	*x* _0_	*x* _1_	*x* _2_	*x* _3_	*y* _0_	*y* _1_	*y* _2_	*y* _3_	*y* _4_
States	0	1	2	3	0	1	2	3	4
Average stay	1.0	1.2	1.5	2 0.5	2.0	2.0	2.0	2.0	2.0
Output density λ	1.0	0.83	0.66	0.4	0.5	0.5	0.5	0.5	0.5

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
