# Peer review of "Model of Random Field with Piece-Constant Values and Sampling-Restoration Algorithm of Its Realizations"

_entropy, 2019, doi:10.3390/e21080792_

Round 1

Reviewer 1 Report

The term SRA should be defined in the first time that is presented

Some equations should be improved, there are symbols not clear.

Why do the authors use the function z=x+y?

Reviewer 2 Report

A description of the model of a random piecewise constant field formed by the sum of realizations of two Markov processes with an arbitrary number of states and defined along mutually perpendicular axes is proposed.

Please explain the principal contributions of this work; it is seemed that there is presented the repetition of results from ref. 23 and other authors papers in this part of the paper (Sect. 4). Please explain the principal scientific contribution of this current paper and the differences with previously paper published by authors.

 Two examples explained have only academic interest and this reviewer did find any application for image processing problem mentioned in introduction.

 In the manuscript, there are a lot places (eqs. 1, 2, 3, 12, 13, 19, etc.) where in the equations, the  sub-indexes and parentheses are presented in non-correct form. Please revise and correct theses parts of the equations.  Several equations are written in such  form that potential reader could not understand text of the manuscript.

In addition, this review saw that sizes of letters in different equations are changed, this give additional difficulties in reading of the text.

Small grammar and stylistic errors should be corrected, mainly in commas.

Round 2

Reviewer 2 Report

Revise line 50, page 2; page 14, lines 332 and 339. There are small errors